# Doppler Ultrasonographic Assessment of Abdominal Aortic Flow to Evaluate the Hemodynamic Relevance of Left-to-Right Shunting Patent Ductus Arteriosus in Dogs

**DOI:** 10.3390/ani14101404

**Published:** 2024-05-08

**Authors:** Anne E. van de Watering, Sophie A. M. van Rossem, Marco Baron Toaldo, Niek J. Beijerink, Alma H. Hulsman, Viktor Szatmári, Giorgia Santarelli

**Affiliations:** 1Department of Clinical Sciences, Faculty of Veterinary Medicine, University of Utrecht, 3584 CS Utrecht, The Netherlandsv.szatmari@uu.nl (V.S.); g.santarelli@uu.nl (G.S.); 2Veterinaire Specialisten Vught, Reutseplein 3, 5264 PN Vught, The Netherlands; 3Division of Cardiology, Clinic for Small Animal Medicine, Vetsuisse Faculty, University of Zurich, 8006 Zürich, Switzerland

**Keywords:** blood flow pattern, echocardiography, antegrade, absent, retrograde, spectral doppler, vascular, congenital heart disease

## Abstract

**Simple Summary:**

Patent ductus arteriosus is a common congenital heart defect affecting various species, including dogs and humans. It can cause congestive heart failure and death if left-to-right shunting is substantial. In humans, assessment of the hemodynamic relevance of a patent ductus arteriosus includes evaluation of the post-ductal aortic flow pattern. We found that ultrasonographic assessment of abdominal aortic flow was feasible in dogs with a patent ductus arteriosus. However, in dogs with higher heart rates, assessment of end-diastolic flow was not always possible. A retrograde end-diastolic flow was the most accurate parameter for identifying dogs with a hemodynamically significant patent ductus arteriosus.

**Abstract:**

In this multicenter, prospective, observational study, abdominal aortic flow was examined with pulsed-wave Doppler ultrasound in dogs with a left-to-right shunting patent ductus arteriosus (PDA) and in apparently healthy dogs. Forty-eight dogs with a PDA and 35 controls were included. In the dogs with a PDA, 37/48 had hemodynamically significant PDAs (hsPDAs) while 11/48 had non-hsPDAs, based on the presence or absence of echocardiographic signs of left-sided volume overload, respectively. In 12 dogs (4/35 control dogs, 7/37 dogs with an hsPDA and 1/11 dogs with a non-hsPDA), the diastole was too short to visualize the end-diastolic flow. Antegrade end-diastolic flow was observed in 30/35 controls and 6/11 dogs with a non-hsPDA. Absent end-diastolic flow was observed in 1/35 control dogs and 3/11 dogs with a non-hsPDA. Retrograde end-diastolic flow was observed in 30/37 dogs with an hsPDA and 1/11 dogs with a non-hsPDA. Twenty-one dogs (15 with an hsPDA and 6 with a non-hsPDA) were reassessed after PDA closure, and, in 19/21, end-diastolic flow was visualized: 17/19 showed an antegrade flow, 1/19 an absent flow and 1/19 a retrograde flow. Sensitivity and specificity of retrograde end-diastolic flow for detection of hsPDAs were 100% and 90%, respectively. In conclusion, ultrasonographic assessment of abdominal aortic flow was feasible in dogs with PDA. However, end-diastolic flow was not always visualized. The presence of a retrograde end-diastolic flow was an accurate finding for discriminating hsPDAs and non-hsPDAs.

## 1. Introduction

A patent ductus arteriosus (PDA) is one of the most common congenital heart defects in various species, including dogs and humans [1,2]. If the amount of blood shunting through the ductus is substantial, it can lead to left-sided congestive heart failure, pulmonary hypertension and death [3]. In dogs, not having a PDA closed negatively affects survival, with a reported mortality rate of 64% during the first year of life [4,5]. Therefore, PDA closure by means of surgical or catheter-based techniques is usually advised. These procedures are performed routinely in specialist referral veterinary practices with low morbidity and mortality, but costs are usually high, and procedural-associated risks and post-operative complications remain present [6,7]. Hence, intervening on an asymptomatic puppy should have a sound indication to prevent the development of cardiac-related clinical signs and death.

In human medicine, it is suggested that only hemodynamically significant PDAs (hsPDAs) should be closed, but there is no consensus on what exactly defines the defect as hemodynamically significant [8,9]. In the absence of clinical signs, echocardiography is used to assess the characteristics of the ductus, the presence of signs of pulmonary overcirculation indicated by volume overload of the left heart and the presence of signs of systemic hypoperfusion indicated by abnormal blood flow patterns in the post-ductal aorta and other abdominal arteries [8,10]. More specifically, the diastolic flow in these vessels is expected to be antegrade as the blood flows distally in the presence of a non-hsPDA, as in healthy individuals, while an absent or retrograde diastolic flow would be indicative of an hsPDA [11]. Such a retrograde post-ductal aortic flow, due to the diastolic run-off of blood through the PDA towards the pulmonary circulation, has been shown to have a strong correlation with ductal shunt volume through phase contrast magnetic resonance imaging in humans [11].

To the authors’ knowledge, abdominal aortic flow patterns have not been routinely evaluated in dogs with a PDA. However, this has been carried out in healthy dogs [12]. The purpose of this study was to determine whether abdominal aortic flow investigation by qualitatively evaluating the flow patterns obtained using pulsed-wave Doppler ultrasonography and comparing the spectra to those of dogs without obvious heart disease can contribute to the assessment of the hemodynamic relevance of PDAs in dogs.

## 2. Materials and Methods

In this multicenter, prospective, observational study, privately owned dogs examined at three different veterinary hospitals (Veterinary Teaching Hospital of the University of Utrecht, referral center Veterinaire Specialisten Vught and the Veterinary Teaching Hospital of the University of Zürich) were prospectively enrolled between October 2021 and January 2024, after obtaining their owners’ consent. Institutional ethical approval was not requested as the dogs received the echocardiography as part of their diagnostic work-up. Dogs were either referred for murmur investigation or occlusion of a known PDA, screened for breeding purposes or participated in another study to establish breed-specific reference ranges (ethical approval WP 10813-2023-02).

### 2.1. Animals

The data obtained at the time of enrollment included signalment, history and physical examination findings. The diagnostic work-up included a complete echocardiographic examination and pulsed-wave Doppler ultrasonographic assessment of the abdominal aortic flow. Dogs with a diagnosis of a left-to-right shunting PDA were included in the case group, while apparently healthy dogs without echocardiographic abnormalities were included in the control group. Dogs with concurrent cardiac disease or with aortic valve insufficiency were excluded. Dogs with a PDA were further divided into two subgroups depending on the hemodynamic significance of their defect, as explained in the following section. Dogs with a PDA that underwent ductal closure were re-examined after the procedure, when possible, and this was considered successful with the absence of a murmur and no residual flow on echocardiography.

### 2.2. Echocardiography

All dogs were manually restrained in lateral recumbency and examined without sedation by a board-certified cardiologist or a supervised cardiology resident. Complete standard transthoracic echocardiography was performed according to established methods using 2-dimensional M-mode color and spectral Doppler ultrasonography [13,14]. The left atrial size was assessed by measuring the left-atrial-to-aortic-root ratio (LA:Ao) on 2-dimensional images from the right parasternal short axis view at the end of the ventricular systole [15]. Left ventricular end-diastolic internal dimensions (LVIDd) were measured with M-mode from a right parasternal short axis view at the level of the chordae tendineae and indexed to body weight using allometric scaling (LVIDdN) [16]. End-diastolic left ventricular volume (EDV) was measured with the modified Simpson’s method of discs at end diastole from a right parasternal four-chamber long-axis view and/or left apical four-chamber view [14,16].

In dogs with a PDA, the minimal ductal diameter (pulmonary ostium), shunt direction and peak systolic and end-diastolic blood flow velocities across the ductus were assessed from the left cranial and/or the right parasternal transverse heart base view [3,17]. Aortic peak flow velocity was obtained from the subcostal view [18].

Dogs with a PDA were included in the subgroup of hsPDA in the presence of one or more of the following findings indicating pulmonary overcirculation: LA:Ao > 1.6 [19], LVIDdN > 1.85 [16], EDV > 3.27 mL/kg from right parasternal four-chamber view or >3.21 mL/kg from left apical four-chamber view [20]. In the absence of these abnormalities, dogs were included in the subgroup of non-hsPDA.

### 2.3. Doppler Ultrasonographic Assessment of Abdominal Aortic Flow

Abdominal aortic flow was obtained in all dogs in right lateral recumbency directly after completion of the echocardiographic assessment from this side. Various high-frequency probes were used for this purpose according to available equipment. The transducer was placed with a sagittal orientation on the caudo–dorsal part of the abdomen, and the abdominal aorta was imaged in a longitudinal section between the aortic trifurcation and the renal artery. A pulsed-wave Doppler mode was used to record abdominal aortic flow with a sample volume that filled the vessel lumen, excluding its walls. No computed angle adjustment tool was used, but the insonation angle between the axis of the ultrasound beam and the axis of the vessel was kept as parallel as possible to the blood flow, as recommended [13]. The wall filter was set to the lowest setting.

The abdominal aortic flow patterns were evaluated qualitatively at a later moment from recorded images by a board-certified cardiologist (GS) and a supervised cardiology resident (AW) who were not blinded to the dog’s allocation in the different groups. Consensus between the two observers was needed for the classification of the different abdominal aortic flow patterns.

In diastole, when the aortic segment caudal to the renal artery is imaged, a triphasic flow pattern is expected in healthy subjects, characterized by an initial negative deflection followed by a positive rebound and a third phase referred to as end-diastolic flow [21]. Based on the direction of the end-diastolic flow, abdominal aortic flow patterns were classified as one the following three types: antegrade when blood flowed caudally at the end of diastole (Figure 1); absent when no blood flow could be detected at the end of diastole (Figure 2); retrograde when blood flowed cranially at the end of diastole (Figure 3) [11].

### 2.4. Statistical Analysis

Statistical analysis was performed with commercially available software (IBM SPSS Statistics 29.0, 2022). Distribution of variables was tested for normality by use of the Shapiro–Wilk test (α = 0.05). Normally distributed data are reported as mean and standard deviation and non-normally distributed data as median and range. Comparisons between groups for continuous variables were carried out using a Student’s independent *t*-test or a Mann–Whitney U test when data were normally and not normally distributed, respectively. Categorical variables were compared between groups by Chi-square test. Sensitivity and specificity of absent and retrograde abdominal aortic flow patterns for detection of hsPDAs were calculated. *p* values < 0.05 were considered statistically significant.

## 3. Results

### 3.1. Study Sample

The initial study sample comprised 93 dogs, of which 10 were excluded due to insufficient quality of abdominal aortic flow imaging. The final study sample comprised 83 dogs, of which 48 were included in the case group and 35 were controls.

The case group comprised 30 intact females, 9 intact males, 6 spayed females and 3 castrated males. The median age of the dogs in this group was 9.5 months (range 2–89 months), and the median body weight was 6.5 kg (range 1–52 kg). Eight dogs were mixed breeds; all other dogs were purebreds of 16 different breeds, the most common being Pomeranian (n = 11) and Poodle (n = 3). Thirty-seven dogs were included in the hsPDA subgroup and 11 dogs in the non-hsPDA subgroup. There was no statistical difference in age, weight and sex between the two subgroups. Three dogs with an hsPDA showed clinical signs compatible with congestive left-sided heart failure, which was supported by compatible findings on thoracic radiographs. No other dogs in any group showed clinical signs. The median murmur intensity in dogs with an hsPDA was 5/6 (range 4–6/6) and in dogs with a non-hsPDA 4/6 (range 2–5/6). Murmur intensity was significantly higher in dogs with an hsPDA (*p* < 0.001) compared to dogs with a non-hsPDA. In twenty-one dogs (15 with an hsPDA and 6 with a non-hsPDA), the abdominal aortic flow could be reassessed post-closure, either on the same day as the procedure when they were fully awake or at the recheck visit 1–3 months later. Nineteen dogs underwent transarterial ductal occlusion using either an Amplatz canine duct occluder (n = 15), a thrombogenic coil (n = 2) or a vascular plug (n = 2) and two had surgical ligation via thoracotomy.

The control group comprised 15 intact females, 10 intact males, 7 spayed females and 3 castrated males. The median age was 36 months (range 9–144 months), and the median body weight was 21 kg (range 4.0–40 kg). Six dogs were mixed breeds; all other dogs were purebreds of eight different breeds, the most common being Labrador Retriever (n = 10) and Dutch sheepdog (n = 10). Control dogs were significantly heavier and older than dogs with PDA (*p* < 0.001). No significant differences were present concerning sex (*p* = 0.905).

### 3.2. Echocardiographic Findings and Abdominal Aortic Flow Pattern

Results for the echocardiographic variables obtained in dogs with a PDA are summarized in Table 1.

In 12 dogs (4/35 control dogs, 7/37 dogs with an hsPDA and 1/11 dogs with a non-hsPDA), the duration of the diastole did not allow visualization and characterization of the end-diastolic aortic flow (Figure 4). Mean heart rate was 110 (±32) beats per minute in dogs with a visible abdominal aortic end-diastolic flow and 134 (±33) beats per minute in dogs in which this could not be visualized; the difference between mean heart rates was significant (*p* = 0.024).

In the rest of the dogs, the following patterns were observed (Figure 5): antegrade (30/35 controls and 6/11 dogs with a non-hsPDA), absent (1/35 controls and 3/11 dogs with a non-hsPDA) and retrograde (30/37 dogs with an hsPDA and 1/11 dogs with a non-hsPDA).

Twenty-one dogs (15 with an hsPDA and 6 with a non-hsPDA) were re-examined after ductal closure. In two dogs, the length of the diastole had precluded pattern characterization. Seventeen dogs showed an antegrade pattern, of which 13/17, 2/17 and 2/17 had shown, respectively, a retrograde, absent or antegrade pattern before closure. One dog with a non-hsPDA showed an absent pattern before and after successful closure. One dog with an hsPDA, assessed after unsuccessful surgical PDA closure, showed a retrograde end-diastolic flow.

Sensitivity and specificity of retrograde end-diastolic flow for detection of an hsPDA were 100% and 90%, respectively. Sensitivity and specificity of absent end-diastolic flow for the detection of an hsPDA were 0% and 70%, respectively.

## 4. Discussion

To the authors’ knowledge, the present study is the first one in which the abdominal aortic flow pattern was investigated in dogs with a PDA. Assessment proved to be feasible, but, in some dogs, flow pattern characterization was precluded by the short duration of the diastole at higher heart rates. When visible, retrograde end-diastolic flow was shown to be an accurate variable for discriminating dogs with an hsPDA and dogs with a non-hsPDA.

The hemodynamic relevance of a PDA depends on the shunting volume passing through the ductus, and therefore on its size, on the systemic and pulmonary vascular resistances and on the adaptability of the left ventricle to accommodate the increased volume load [10]. In humans with a PDA, spectral Doppler flow pattern assessment of the post-ductal aorta, superior mesenteric artery and/or anterior cerebral artery is commonly employed to evaluate the systemic perfusion and, therefore, indirectly, the hemodynamic significance of the PDA [10].

The perfusion requirements of each organ determine the characteristics of the respective vascular bed, which, in turn, influences the flow profile of the associated vessels. Peripheral arterial Doppler waveform description is based on, among other things, the display of the flow components relevant to the zero-flow baseline [22]. In humans, in the absence of disease, the abdominal aortic waveform is described as multiphasic, with an initial systolic sharp upstroke produced by the contraction of the left ventricle and by the consequent high-velocity forward movement of a bolus of blood. The diastolic component will then depend on the level of vasoconstriction present in the resting muscular beds. Due to the high resistivity of the vascular bed of the hind limbs, the aortic Doppler tracing obtained caudally to the renal arteries will show a retrograde wave in early diastole, due to the backward movement of blood. An antegrade component should then be observed in mid-to-late diastole, produced as a result of an antegrade wave generated by proximal compliant large and medium-sized arteries [22]. In the presence of a PDA, a phenomenon called “ductal steal” occurs, for which blood flows from the aorta into the pulmonary artery throughout the cardiac cycle [23]. Due to this phenomenon, when left-to-right shunting of blood is substantial, the diastolic antegrade flow in the post-ductal aorta can disappear or even become retrograde [23]. Therefore, an absent or retrograde diastolic flow recorded in this vessel would suggest systemic hypoperfusion and therefore be considered indicative of an hsPDA [8].

Based on the results of this study, abdominal aortic flow assessment could also be useful in dogs to help evaluate the hemodynamic relevance of a PDA and, therefore, the need for surgical closure. At present, as a consequence of the poor survival rate reported, closure of a PDA in dogs is generally recommended in the veterinary literature [24,25,26,27]. However, there are occasional reports of dogs in which closure is not attempted due to them having small PDAs considered to be hemodynamically not significant [5]. Therefore, while the majority of PDAs in dogs might require closure, it is possible to encounter a minority of patients in which the benefits of an intervention are less clear. It must be remembered that even minimally invasive procedures are not without risks, with various complications reported to occur, such as new-onset arrhythmias, hemorrhage from vascular perforation, embolization of the occlusion device, cardiac arrest and bacterial ductal arteritis [5,6,26,28]. Evaluation of the abdominal aortic flow could therefore help establish the hemodynamic significance of a PDA and the need for treatment. This could be especially valuable in the absence of secondary cardiac remodeling. In fact, the dilation of the left atrium and ventricle as a response to the volume overload is reported as gradual and progressive and the time of exposure as a factor influencing the evaluation of the hemodynamic significance of a PDA [29,30]. In the present study, a retrograde pattern was observed in a 3-month-old dog included in the non-hsPDA group, suggesting systemic hypoperfusion and possible misclassification if measurable cardiac remodeling had not occurred yet at the time of presentation. Follow-up studies in dogs that did not undergo closure could help establish whether a retrograde flow in absence of cardiac remodeling is predictive of future cardiomegaly.

Abdominal aortic flow assessment could also help to evaluate residual flow in dogs that undergo PDA closure. In this study, a retrograde end-diastolic flow was observed in one dog that underwent incomplete surgical closure due to a major intra-operative complication, while most of the dogs re-evaluated after successful closure showed an antegrade aortic flow pattern.

Based on the results of this study, another variable that could contribute to the evaluation of the hemodynamic significance of a PDA is murmur intensity, even though overlapping between the two groups was observed. Low-grade cardiac murmurs suggest non-hsPDAs. The positive correlation between murmur intensity and disease severity has also been described in dogs with myxomatous mitral valve disease and pulmonic and subaortic stenosis [31,32]. Another helpful non-echocardiographic variable could be the N-terminal pro-B-type natriuretic peptide. However, levels of this molecule were not determined in the present study as biological variability affects its concentrations in dogs [33] and additional blood sampling would have been necessary. Furthermore, it is a variable that reflects the presence of cardiac remodeling in response to volume overload of the left heart, which could be lacking in early stages even in the presence of substantial shunting.

In contrast to what has been reported in humans, for which the presence of an absent end-diastolic flow in the descending aorta would suggest a moderately large hsPDA [8], in the present study, the absent pattern was also observed in both apparently healthy dogs and dogs with a non-hsPDA. In dogs with a non-hsPDA, an absent pattern could still indicate hemodynamic significance if misclassification had occurred due to absence of cardiac remodeling in very young dogs. However, this finding in control dogs is unexpected and could reflect a peculiarity of the species, technical issues or misclassification. Malalignment could influence the spectral display of the abdominal aortic waveform, creating falsely absent patterns; however, the insonation angle between the axis of the ultrasound beam and the axis of the vessel was kept as parallel as possible to the blood flow, as recommended [13,34]. Control dogs had no obvious clinical signs or demonstrable abnormalities in physical examination or echocardiography, but the presence of extra-cardiac left-to-right shunts cannot be completely excluded.

In humans, ductal characteristics, including size, are also evaluated to assess the hemodynamic significance of a PDA. Ductal size was recorded in this study but excluded from statistical analysis because the pulmonary ostium diameter should have been corrected considering the wide range of body weights. To the authors’ knowledge, such correction has not been standardized yet in dogs. Furthermore, the majority of dogs have an oval-shaped pulmonary ostium, for which transthoracic echocardiographic assessment in a single dimension could lead to unreliable measurements in the same patient depending on the view in which the PDA is imaged and the angle obtained [35].

The present study has several limitations. The investigators in this study were not blinded to the history, physical examination findings, echocardiography results and surgical outcomes; therefore, observer bias cannot be excluded. However, to decrease such bias, consensus between two observers was needed for the classification of the different abdominal aortic flow patterns. A second limitation is the classification of dogs in the group of non-hsPDAs based on the absence of echocardiographic signs of volume overload as diagnosis at a very young age could have resulted in misclassification. Furthermore, artificially low/normal left cardiac dimensions related to the presence of a patent foramen ovale or an atrial septal defect cannot be completely excluded as these defects could be missed on a standard echocardiogram [29]. Obtainment of the echocardiogram-estimated pulmonary-to-systemic-flow ratio could have helped to classify dogs with a PDA more accurately, but this variable was not included in the protocol as performing its measurement adequately is known to be challenging in dogs with a PDA [36]. Lastly, the case group and control group were not matched based on age and body weight, and the association between these variables and the abdominal aortic flow pattern was not examined. While heart rate in dogs undergoing routine clinical examination seems to be related to age, with dogs younger than 12 months having higher heart rates, this does not appear to be related to body weight [37].

## 5. Conclusions

In conclusion, non-invasive spectral Doppler flow pattern assessment of abdominal aortic flow was feasible in dogs with a PDA. In some dogs, assessment of end-diastolic flow was precluded by an insufficient duration of the diastole at higher heart rates. An antegrade end-diastolic flow direction was observed in control dogs, dogs with a non-hsPDA and dogs after successful PDA closure. A retrograde end-diastolic flow was mainly observed in dogs with an hsPDA or after unsuccessful ductal closure and allowed accurate discrimination of hsPDAs and non-hsPDAs. An absent end-diastolic flow could be observed in various groups and lacked diagnostic value.

## Figures and Tables

**Figure 1 animals-14-01404-f001:**
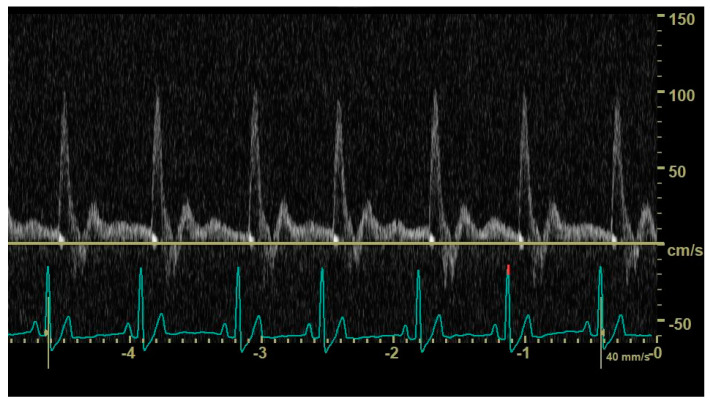
Pulsed_wave Doppler ultrasonography of abdominal aortic flow in a healthy dog showing an antegrade end-diastolic flow.

**Figure 2 animals-14-01404-f002:**
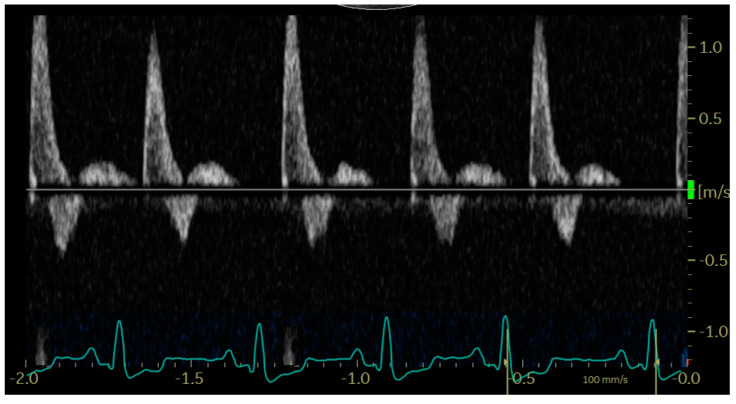
Pulsed_wave Doppler ultrasonography of abdominal aortic flow in a dog with left-to-right shunting patent ductus arteriosus showing an absent end-diastolic flow, with the exception of the first cardiac cycle, where the diastole is too short to allow its visualization.

**Figure 3 animals-14-01404-f003:**
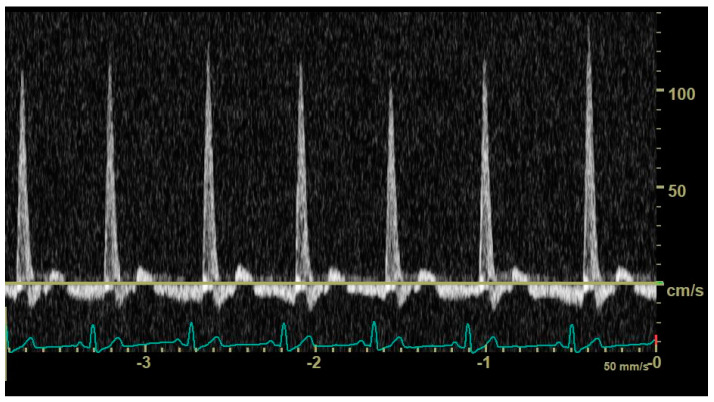
Pulsed_wave Doppler ultrasonography of abdominal aortic flow in a dog with left-to-right patent ductus arteriosus showing a retrograde end-diastolic flow.

**Figure 4 animals-14-01404-f004:**
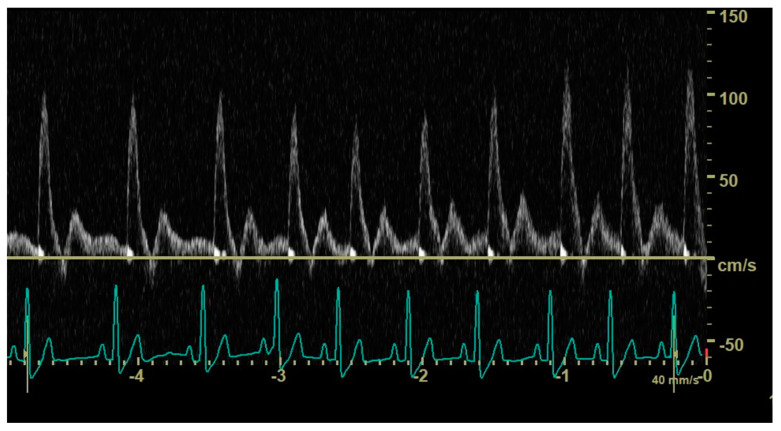
Pulsed_wave Doppler ultrasonography of abdominal aortic flow in a healthy dog showing the effect of an increasing heart rate on the end-diastolic flow; in longer cycles, an antegrade end-diastolic flow can be observed, while, in shorter cycles, visualization and characterization of end-diastolic flow are not possible.

**Figure 5 animals-14-01404-f005:**
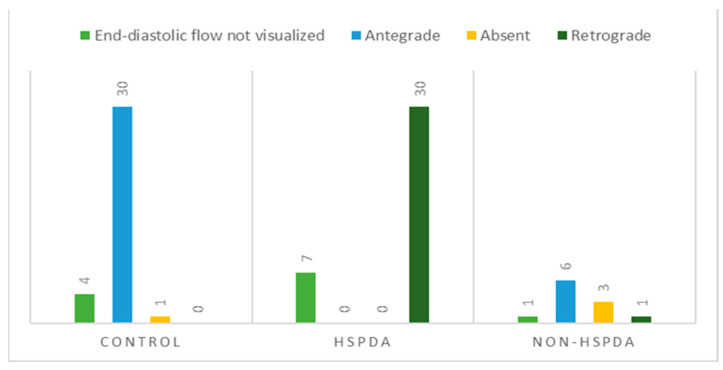
Distribution of abdominal aortic flow patterns based on evaluation of end-diastolic flow direction in control dogs, dogs with hemodynamically significant patent ductus arteriosus (hsPDA) and dogs with non-hemodynamically significant patent ductus arteriosus (non-hsPDA).

**Table 1 animals-14-01404-t001:** Echocardiographic variables obtained in 48 dogs with a left-to-right shunting patent ductus arteriosus. Values are expressed as mean and standard deviation if normally distributed and as median and range if not normally distributed.

	non-hsPDA (n = 11)	hsPDA (n = 37)
LA:Ao	1.28 (±0.17)	1.53 (±0.28) *
LVIDdN (cm/kg)	1.57 (±0.12)	2.30 (±0.35) *
EDV (mL/kg)	1.75 (1.33–2.80)	6.25 (2.65–13.07) *
Minimal ductal diameter	1.96 (±0.72)	2.50 (±0.93)
PDA peak systolic velocity (m/s)	5.06 (±0.53)	4.90 (±0.52)
PDA end-diastolic velocity (m/s)	3.94 (3.1–4.7)	3.33 (1.5–3.97) *
Aortic flow velocity (m/s)	1.61 (±0.38)	2.20 (±0.46) *

* Within a row, value differs significantly (*p* < 0.01). Hs = hemodynamically significant; non-hs = non-hemodynamically significant; LA:Ao = left-atrial-to-aortic-root ratio; LVIDdN = left ventricular end-diastolic internal diameter normalized to body weight; EDV = end-diastolic volume; PDA = patent ductus arteriosus.

## Data Availability

Data are contained within the article.

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
