# Peer review of "Doppler Ultrasonographic Assessment of Abdominal Aortic Flow to Evaluate the Hemodynamic Relevance of Left-to-Right Shunting Patent Ductus Arteriosus in Dogs"

_animals, 2024, doi:10.3390/ani14101404_

Round 1
Reviewer 1 Report
Comments and Suggestions for Authors
Thank you for submitting this interesting paper. In the attached PDF I left comments related to the highlighted parts within the text.

Author Response
We appreciate the careful review of the manuscript
Minor Comments:
Line 28, 33, 129, 135, 181, 214, 226, 300 and 347
The changes suggested have been made.
Line 112 and 334: This is not relevant to the study as its purpose is to evaluate abdominal aortic flow in hs VS non-hsPDA. You use routine echocardiographic variable to determine whether it is hs or non-hs (left volume overload or left CHF
This information has been removed.
Line 175: Where these 2 groups different in age/weight/sex?
The statistic analysis to answer to this question has been done and the results have been added to the manuscript.
Line 207: I understand the main discriminating factor between hs and non-hs from abdominal aortic flow is the end-diastolic flow (anterograde, absent, retrograde).
If we do not have the end-diastolic flows of these dogs, why were they included?
We thank the Reviewer for raising this important issue. This phenomenon has not been described in humans, and seen its prevalence in our group of dogs, we consider important to include them and make the clinicians aware of the fact that assessment of abdominal aortic flow might be inconclusive in some dogs.
Line 217: Was this image taken from a right recumbent animal?
This has been described in the Materials and Methods section.
Line 219: What is the explanation for an absent diastolic flow in a control dog?
This has been described in the Discussion section.
Line 259: Due to the high resistivity of the vascular bed of the hind limbs, the aortic Doppler tracing obtained caudally to the renal arteries will show a retrograde wave in early diastole, due to the backward movement of blood.
We thank the Reviewer for bringing this to our attention and we have modified the text accordingly.
Line 140 If we consider this, then the whole value of study might be questionable since it is based on end-diastolic abdominal aortic flow.
This comment is unclear to us, the sentence is only related to the absent pattern and not to end-diastolic flow in general.
Reviewer 2 Report
Comments and Suggestions for Authors
Patent ductus arteriosus (PDA) is a common congenital heart defect affecting both dogs and humans. The defect may cause congestive heart failure and death if left-to-right shunting is present. In humans, the hemodynamic evaluation of PDA is performed by the assessment of the post-ductal aortic flow pattern. This is applicable in dogs, except when high heart rates are present. A retrograde end-diastolic flow was the most accurate parameter to identify dogs with a hemodynamically significant patent ductus arteriosus. This study aimed to determine whether abdominal aortic flow investigation can contribute to the assessment of the hemodynamic relevance of PDA in dogs, by evaluating the flow patterns obtained using pulsed-wave Doppler ultrasonography and comparing with dogs not affected by cardiac disease or defect. The paper is well written, the study design is appropriate, and the results are clearly presented and discussed.
Comments on the Quality of English LanguageMinor editing of the English language is required.
Author Response
We thank the Reviewer for this positive evaluation of our manuscript.
Reviewer 3 Report
Comments and Suggestions for Authors
REVIEW: animals 2975844
Article: Doppler ultrasonographic assessment of abdominal aortic flow to evaluate the hemodynamic relevance of left-to-right shunting patent ductus arteriosus in dogs
General comments:
This is an interesting study of which the post-ductal aortic flow pattern was assessed in dogs with left-to-right shunting PDA in order to determine whether it can be used to discriminate the hsPDA to non-hsPDA. As it has already been used in human for the assessment of the severity of PDA, I believe that this is a very promising parameter for dogs as well.
While I think the design of the study is well thought out, I am also concerned about the control group selected for this study. As looking at the results of the study, I have noticed a big discrepancy between the case group and control group regarding the age and body weight, and I do not think that the influence of age and body size on the results cannot be ignored.
I have also noticed that the echocardiographic parameters such as FS and mitral E-velocity were not included in the analysis. Please consider adding these parameters as I think it will give an additional value to the study.
Specific comments:
Line 25: Please specify what you mean by the “In 12 dogs”
Lin 28: Are these “twenty-one dogs” classified as hs-PDA?
Line 93: Would it be possible to state the timing of the re-examination?
Line 177: Please specify the clinical signs observed. Also, where any of these dogs on medications?
Line 226-233: The results presented regarding the abdominal aortic flow after the ductal close are extremely difficult to follow. Please consider showing the data in a different manner, for example with the use of tables, figures, etc.
Line 317: Please discuss whether there was any association between age or the size of the dogs on the pattern of the abdominal aortic flow. For instance, older and bigger dogs usually have a slower heart rate, that may have an effect.
Line 345-357: Please add the age and body weight differences between the groups as limitation.
Author Response
We appreciate the careful review of the manuscript
Article: Doppler ultrasonographic assessment of abdominal aortic flow to evaluate the hemodynamic relevance of left-to-right shunting patent ductus arteriosus in dogs
General comments:
This is an interesting study of which the post-ductal aortic flow pattern was assessed in dogs with left-to-right shunting PDA in order to determine whether it can be used to discriminate the hsPDA to non-hsPDA. As it has already been used in human for the assessment of the severity of PDA, I believe that this is a very promising parameter for dogs as well.
We thank the Reviewer for this positive comment.
While I think the design of the study is well thought out, I am also concerned about the control group selected for this study. As looking at the results of the study, I have noticed a big discrepancy between the case group and control group regarding the age and body weight, and I do not think that the influence of age and body size on the results cannot be ignored.
We thank the Reviewer for raising this important issue and have expanded the Limitations section accordingly.
I have also noticed that the echocardiographic parameters such as FS and mitral E-velocity were not included in the analysis. Please consider adding these parameters as I think it will give an additional value to the study.
We agree with the comment of the Reviewer. However, while the mitral E/A ratio is easy to measure, this parameter is not consistently used in humans to evaluate the hemodynamic significance of a PDA as in preterm infants, the E/A ratio gradually becomes >1 as the myocardium compliance improves. In fact, in preterm infants, mitral valve E/A ratio is usually <1 due to poor compliance of the myocardium leading to moderate impairment of diastolic performance and low early diastolic filling velocity. In the presence of a hsPDA, the atrial pressure increases because increased pulmonary venous return and this leads to a reversal of the E/A ratio >1 (Ref. 29). To the authors' knowledge, the diastolic performance and myocardial compliance have not been investigated in puppies with PDA. For this reason, the parameters FS and mitral E-velocity were not included in the study.
Specific comments:
Line 25: Please specify what you mean by the “In 12 dogs”
This has been added to the manuscript.
Lin 28: Are these “twenty-one dogs” classified as hs-PDA?
The classification has been added.
Line 93: Would it be possible to state the timing of the re-examination?
The timing has been added, see line 188.
Line 177: Please specify the clinical signs observed. Also, where any of these dogs on medications?
The clinical signs are described in line 181.
Line 226-233: The results presented regarding the abdominal aortic flow after the ductal close are extremely difficult to follow. Please consider showing the data in a different manner, for example with the use of tables, figures, etc.
We thank the Reviewer for bringing this to our attention and we have corrected and rewrote this part.
Line 317: Please discuss whether there was any association between age or the size of the dogs on the pattern of the abdominal aortic flow. For instance, older and bigger dogs usually have a slower heart rate, that may have an effect.
We have altered the text and included a new reference.
Line 345-357: Please add the age and body weight differences between the groups as limitation.
This has been added to the manuscript.